`ProteinShake`
# Building datasets and benchmarks for deep learning on protein structures

**Tim Kucera** [1,2,3,*], **Carlos Oliver** [1,2,3 *], **Dexiong Chen** [1,2,3], and **Karsten Borgwardt** [1,2,3]

[1]Department of Biosystems Science and Engineering, ETH Zürich, Switzerland
[2]SIB, Swiss Institute of Bioinformatics, Basel, Switzerland
[3]Max Planck Institute of Biochemistry, Martinsried, Germany
[*]These authors contributed equally.
{kucera, oliver, dchen, borgwardt}@biochem.mpg.de

## Abstract

We present `ProteinShake`, a Python software package that simplifies dataset creation and model evaluation for deep learning on protein structures. Users can create custom datasets or load an extensive set of pre-processed datasets from biological data repositories such as the Protein Data Bank (PDB) and AlphaFoldDB. Each dataset is associated with prediction tasks and evaluation functions covering a broad array of biological challenges. A benchmark on these tasks shows that pre-training almost always improves performance, the optimal data modality (graphs, voxel grids, or point clouds) is task-dependent, and models struggle to generalize to new structures. `ProteinShake` makes protein structure data easily accessible and comparison among models straightforward, providing challenging benchmark settings with real-world implications.
`ProteinShake` is available at https://proteinshake.ai.

## 1 Introduction

Over the decades, data describing protein sequence, structure, and function has been amassed in databases such as UniProt [13] and RCSB PDB [6], as well as in a plethora of domain-specific databases [5, 38, 12]. This abundance of data presents a prime application for deep learning models and an opportunity to generate new insights with machine learning in challenging real-world scenarios. The recent addition of millions of high quality structure predictions from AlphaFold [30] and other structure prediction algorithms [32] adds a valuable resource to this pool of data, which researchers are now eager to utilize.

However, the heterogeneity of protein data constitutes a problem for reproducible machine learning. Structures are stored in heavy formats not developed for deep learning models, metadata is often scattered across different databases, and files can be corrupted or of poor quality. Additionally, the diverse approaches and network architectures in geometric deep learning require different representations of the protein structure, such as point clouds, graphs, or voxel grids. This results in highly variable data processing steps and evaluation schemes across publications, and eventually prevents comparison.

Ideally, datasets and evaluations should be standardized. This includes well-defined and transparent pre-processing steps of the data and reproducible data splits in the evaluation, with selected, domain-appropriate metrics. As modern deep learning models tend to be generalists (i.e. operate in multiple contexts and tasks) [45], the data curation process should be harmonized across differ-

ent biological tasks, such that model architectures can be assessed by how well they capture the underlying fundamental biological concepts. Such efforts have already propelled research across different disciplines (e.g. ImageNet [14], OGB [22]), including biological domains (e.g. TAPE [44], MoleculeNet [55], ProteinNet [3], rnaglib [34]). A similar effect can be expected in the recently expanding domain of structural biology data.

We therefore propose ProteinShake to harmonize the pre-processing and evaluation steps of protein structure data from various databases, converting them to deep-learning-ready formats while supporting all major deep learning frameworks and architectures. Protein structures are natively available in different representations such as point clouds, voxel grids, and graphs. For evaluation, we provide standardized data splits based on sequence and structure similarity, with appropriate metrics. The tasks span a wide variety of biologically important prediction targets and machine learning problems, ranging from functional label classification, over pairwise interaction prediction and structural similarity search, to regression on ligand affinity.

The library is easily extended to include (and explicitly designed for) user-contributed datasets and tasks. The source code and our release scripts are fully open-source and regularly updated, such that datasets remain up-to-date and can be reproduced anywhere. All datasets and tasks are hosted in pre-processed form on a public dataset repository to speed up model development. Additionally, we host a public leaderboard [1] as an opportunity for researchers to compare their models.

Our design goal is to provide a simple user interface that abstracts away boilerplate code and removes the need to make decisions that require expert domain knowledge, while maintaining full reproducibility and customizability. We anticipate that ProteinShake will be a useful resource for researchers working at the intersection of machine learning and structural biology, and that it will serve them as a basis for reproducibility and comparability.

## 2  Related work

Previous works have tackled various aspects of the data preparation workflow for protein structures. ProteinShake mainly distinguishes from these works through its *accessibility*, culminating in single-line usage of datasets and evaluators.

We required a number of features for our library under which we will also briefly assess the related work: **Amenability.** Users with all backgrounds should be able to use it without worrying about domain-specific parameters that require expert biological knowledge. **Availability.** All data and splits should be fully reproducible on any machine, without relying on a central database. **Transparency.** The processing steps from raw data to the final data objects should be open-source and self-contained. **Extendability.** Based on the previous points, it should be easy to extend the database with new datasets and tasks. **Compatibility.** The data should be compatible with all popular geometric deep learning approaches. **Recency.** Finally, the library should provide the means to update datasets with newly available data, which requires data versioning.

One of the first efforts was ATOM3D [50], which proposed six biologically relevant protein tasks and provided domain-specific splits for each. The work is accompanied by a Python package which facilitates access to these tasks, and allows users to work with their own data. Datasets imported to ATOM3D are endowed with several utilities such as saving, fetching, splitting, transforms and filters. However, because the library only accepts pre-built datasets, this leaves the challenge of collecting, parsing and curating to the user, thereby limiting amenability, availability, and transparency. Notably, datasets large enough to use for pre-training are not included, although pre-training is becoming the basis for state-of-the-art models recently [45].

TorchDrug [61], a machine learning platform originally developed for drug discovery, is recently expanding towards protein data. The API exposes several pre-built datasets and provides highly tunable and composable processing blocks for custom datasets, with a focus on graph representations of proteins. However, obtaining a ready-to-use dataset object requires composing a series of processing steps (amenability). Further, the processing logic used to derive the datasets from primary data is not public, limiting availability, transparency and extendability.

---

[1] https://proteinshake.ai/#leaderboard

`Graphein` [25] is primarily a library for converting protein structure files into graphs. It provides many customizations and utility functions, and is currently expanding to provide pre-processed datasets (without evaluations). The focus of this library however lies on the graph conversion, leaving many of our requirements unapproached. Similar to `Graphein`, `PyUUL` [41], converts protein structure files to voxel grids and point clouds. `biopandas` [46] and `biopython` [11] perform some fundamental processing on structure files such as parsing and cleaning.

As none of these works have fully addressed the requirements we had as machine learners, we decided to create `ProteinShake` with a focus on simplicity of use. The following sections detail the architecture of the framework and the datasets and tasks we provide in the current release.

## 3    Contribution

`ProteinShake` manages the full data and evaluation workflow, allowing the user to simply "plug in" a model (Figure 1). The library provides datasets and associated evaluation tasks for benchmarking models. Each dataset object contains the logic for every processing step, starting at the download of raw data from primary resources (availability and transparency). Parameters are customizable, but set with appropriate defaults, such that data can be used out of the box (amenability). `ProteinShake` also converts data for all common deep learning frameworks and protein structure representations (compatibility). Task and dataset interfaces are easily customizable such that users can extend elements unique to their application (extendability). The current release hosts a database of eight annotated datasets (Table 1), a large-scale pre-training dataset based on AlphaFoldDB, and ten prediction tasks. The releases are versioned and allow for continuous updates of the datasets (recency). Additionally, we demonstrate the use of ProteinShake by conducting experiments to assess the importance of three key features: protein representation, pre-training, and data splitting.

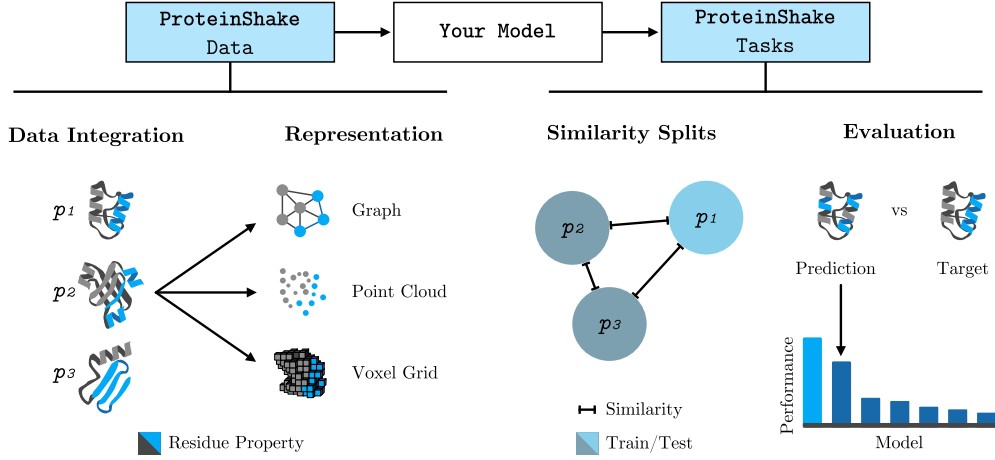

Figure 1: `ProteinShake` simplifies deep learning structural biology workflows from data preparation to model evaluation. We offer annotated datasets which can be converted to several representations (point clouds, graphs, voxel grids) and then cast to several popular computation frameworks (PyTorch, TensorFlow, NumPy/JAX, Pytorch Geometric, DGL, NetworkX). Datasets also serve as the basis of prediction tasks for which we provide domain-specific splits and evaluation metrics.

## 4    Building datasets

A core class in `ProteinShake` is the `Dataset`. It holds a collection of protein objects containing the structure ($x, y, z$ coordinates and atom/residue identities) and associated annotations at the atom, residue, and protein level (*e.g.*, binding status of a residue or functional class of a protein). The protein objects also accommodate meta-information, including quality-related features such as B-factor or pLDDT values. Other relevant features can be added easily, as the protein object is implemented as a flexible, dictionary-type storage.

Each dataset is constructed by implementing three steps: downloading raw data from a source database, filtering according to quality measures, and gathering annotations (possibly from other sources). All surrounding processing work such as API querying, parsing, cleaning, encoding, and storing are taken care of by the library.

Currently, `ProteinShake` provides annotated datasets from eight different areas of protein biology (Table 1) for which we compile ten **supervised** model evaluation tasks (Section 5). We provide two large-scale (unannotated) datasets collected from the RCSB databank and AlphaFoldDB which are primarily intended for **pre-training** and **self-supervised learning**. The former contains a large set of experimentally solved protein structures from the RCSB Protein Data Bank [6] (ca. 36,000 structures), the latter contains computational predictions for several organisms as well as the SwissProt database (ca. 500,000 structures) [30, 7].

Table 1: Datasets currently hosted by `ProteinShake`. The library is designed to be straightforwardly extended, such that new datasets can be easily integrated in the future.

| Dataset | Protein count | Area of protein biology |
|---|---|---|
| Pfam | 31'109 | Evolutionary relationships |
| Gene Ontology | 32'633 | Functions, Components, Pathways |
| Enzyme Commission | 15'603 | Reaction catalysis |
| Ligand binding | 4'642 | Small molecule binding |
| Protein-protein interfaces | 2'839 | Protein binding |
| Structure similarity | 1'000 | Structure alignment |
| Structural class | 10'066 | Geometric relationships |
| Virtual screening | 38 | Drug discovery |
| RCSB monomers | 36'576 | Experimental structures |
| AlphaFold monomers | 541'143 | Predicted structures |

## 4.1 Representations and frameworks

A crucial decision when developing a deep learning model for protein structures is the choice of an appropriate *representation* of the structure which maps raw coordinates and annotations to objects that can be used in the model. This mapping can be done in several ways, each of which have biological implications and also dictate the type of model to be used.

At its core, a protein structure is simply a set of coordinates in 3D space (a **point cloud**). This representation abstracts away the fact that each of the points (atoms or residues) physically interact with each other in a distinct way. One can represent these interactions by constructing a **geometric graph**, where each edge represents a likely interaction as given by spatial proximity. Lastly, one might prefer a more coarse-grained representation where the protein structure is rasterized on a regular **voxel grid**. Each of these representations is rooted in a different field of machine learning and is typically used in dedicated model architectures.

Apart from the general representation, an important nuance of a protein structure is its **surface**, *i.e.*, those atoms and residues that lie on the interface to the solvent and directly interact with the environment. The surface carries special significance for the protein function, as it constitutes the direct interface in binding events. This information can be utilized in modeling.

With this in mind, one of `ProteinShake`'s core features is to allow easy conversions between commonly used protein representations. `ProteinShake` currently supports converting any protein dataset to $\epsilon$-neighborhood or $k$-nearest-neighbor graphs based on inter-atomic or inter-residue distances, voxel grids which represent the occupancy of a regular 3D grid that is laid over the protein structure, and point clouds of atoms or residues which are sets of labeled points in 3D space. Irrespective of the representation, all proteins are annotated with their solvent accessibility scores (computed with `freesasa` [36, 37]), which can either be used as a feature, or as a filter to reduce the protein to its exposed surface residues.

The representations are available for the major deep learning frameworks PyTorch [42], TensorFlow [1], NumPy/JAX [8], and the graph learning frameworks PyTorch-Geometric [16], DGL [53], and NetworkX [21].

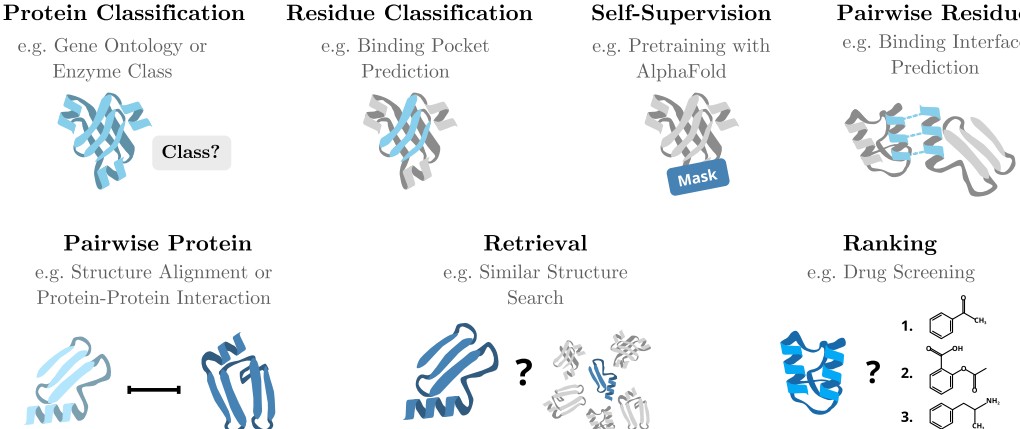

Figure 2: Overview of the supported prediction tasks in the current release. `ProteinShake` is able to model a wide range of biological research questions and machine learning problems, covering regression, ranking, retrieval, classification, pre-training, metric learning, and more. `ProteinShake` is designed to drastically simplify the addition of new tasks, such that new research questions can be addressed quickly.

## 5   Building benchmarks

The second core object in the library is the `Task` which is the central utility for benchmarking. A task extends a dataset with the logic for splitting data into train/validation/test splits and metrics to evaluate a set of predictions. We created and host several tasks based on our annotated datasets which we describe in detail below and illustrate in Figure 2.

### 5.1   Sequence and structure dataset splits

A commonly employed splitting method for protein data is a **sequence-similarity split**. Proteins are first clustered by sequence similarity (*e.g.*, with `CD-Hit` [17]) and the splits are then based on the clusters, ensuring similar instances end up in the same split. This ensures that model generalization can properly be assessed.

It is however known that protein structures are more conserved than protein sequences [24], which implies that dissimilar sequences might still adopt similar structures. Hence, a sequence-based split does not guarantee *structural* dissimilarity between train and test data.

We create a **structure-similarity split** by utilizing the fast structure alignment software `Foldseek` [51]. A complete clustering is computationally prohibitive so we resort to an efficient heuristic: test and validation sets are built iteratively by randomly sampling an initial protein and then retrieving all similar instances with `Foldseek`, constructing structure-based clusters analogously to the sequence-based split. Structural similarity is defined by thresholds on the Local Distance Difference Test (LDDT) metric. Clusters are then sampled until the desired set size is reached. This split procedure serves as a novel and challenging setting that tests the ability of models to generalize in terms of structure and not just sequence homology.

All tasks are available with random, sequence, and structure splits, providing different levels of difficulty for generalization. The splits are pre-computed with similarity thresholds between 30%-90% for the sequence split and 50%-90% for the structure split, with a default of 70%. Users who wish more fine-grained control over the similarity splits can change the parameters when processing locally. `ProteinShake` also offers the ability to add custom split procedures, for example to integrate existing benchmarks that follow other splitting strategies.

### 5.2   Tasks

To showcase the advantage of standardization and the ability of `ProteinShake` to integrate various types of prediction problems, we implement a range of tasks covering different application areas

of biology which we categorize in three major areas: function prediction, geometric reasoning, and physical interaction modeling (Figure 2).

The tasks cover various machine learning settings: regression, multi-class and multi-label classification, retrieval, ranking, metric learning and self-supervised learning. The inputs to the model can be single proteins, pairs of proteins, or proteins paired with other instances such as chemical compounds. Predictions can be made on the atom, residue, or protein level. For each task, several metrics are available, and we designate one metric as the default according to literature where possible.

### 5.2.1 Structure-function relationships

There exist large, expertly curated ontologies that organize proteins into hierarchies of functional roles [5, 12, 4, 13]. In an effort to better understand the relationship between protein structure and the protein's role(s) in the cell we build three classification tasks. In this setting, the input is a protein structure and the prediction target is one of the following classifications:

- *Enzyme Class* (multi-class): given a protein, predict the chemical reaction it catalyzes. Built using the Enzyme Commission hierarchy [4]. The target is given by the top level of the hierarchy. Default metric is accuracy [20].
- *Gene Ontology* (multi-label): given a protein, predict its Gene Ontology [12] labels. The Gene Ontology describes various aspects of molecular function, cellular components, and biological processes. Default metric is Fmax [20].
- *Protein Family* (multi-class): given a protein, predict its protein family (Pfam). The Pfam annotations [5] classify proteins according to the functional attributions of their domains. Default metric is accuracy.

Successful models in these tasks have the potential to deepen our understanding of **structure-function relationships** in proteins [20].

### 5.2.2 Geometric reasoning

Next, we design tasks to test the ability of models to learn geometric relationships between structures. Here we ask whether the learned representations of a model reflect structural properties. We define three prediction targets:

- *Structure Class* (multi-class): given a protein, predict the correct structural class of a protein. This task is is built on the Structural Classification of Proteins (SCOP) [38]. Default metric is accuracy.
- *Structural Similarity* (regression): given an unaligned pair of proteins, predict the (aligned) Local Distance Difference Test (LDDT) of the structures. Target values are computed after alignment with TM-align [58] for all pairs of 1000 randomly sampled single-chain proteins. Default metric is Spearman rank correlation.
- *Structural Search* (retrieval): retrieve a set of proteins structurally similar to a query [51] Similarity is defined by LDDT>=0.8 after alignment with TM-align. Default metric is precision@k.

Models that can learn a good representation of protein structure geometries have applications in fast **structure alignment** [60], efficient **structure search** [51, 33] and **motif discovery** [40].

### 5.2.3 Modeling physical interactions

Physical interactions between proteins and other ligands such as small molecules and RNA form a large part of protein function. The PDBbind-CN [54] database collects and curates a set of experimentally derived protein structures with known binding partners along with additional annotations such as binding affinity and dissociation constants. It has previously been used in binding site prediction and ligand affinity prediction problems [29, 27, 26]. We build three tasks on top of the PDBbind-CN data and the DUDE-Z virtual screening benchmark [47].

- *Ligand Affinity* (regression): given a protein and a chemical compound (represented as a SMILES string), predict the affinity (dissociation constant $K_d$) between the two molecules. Default metric is Pearson correlation [27].

- *Protein Protein Interface* (binary): predict the binary contact matrix between a pair of bound proteins chains. Contact threshold is set to $6.0\,\text{Å}$. Chains are independently centered and randomly rotated to simulate unbound form. Default metric is median AUROC [49, 50], although we also provide metrics for unbalanced data.
- *Binding Site Detection* (binary): given a protein residue, predict whether it belongs to a small molecule binding cavity. Binding site residues are those within the binding pocket provided by PDBBind [54]. Default metric is Matthew's Correlation [18].
- *Virtual Screening* (retrieval): given a protein and a set of small molecules (provided as SMILES strings) containing both active ligands and non-ligands (decoys), rank the small molecules such that actives are ranked higher than the decoys. Default metric is the enrichment factor [10].

Successful models in this family of tasks have potential to accelerate **drug discovery** [52] and to improve our understanding of the **interactome** [57].

## 6 Contributing and maintenance

`ProteinShake` classes handle the boilerplate code of dataset and task creation such that users can implement their custom datasets and tasks by only providing the elements specific to their application. We welcome contributions. See the documentation for more information on how to create datasets and tasks, and how to contribute them.

The preparation of a dataset, even on a compute cluster, can easily take multiple hours up to several days. The pre-processed datasets are therefore hosted[2], providing ready-to-use datasets in the matter of minutes. The `ProteinShake` database has fully automated, versioned releases to incorporate newly added protein structures and annotations. Additionally, the classes contain all processing code, such that they can be built by anyone from scratch, guaranteeing reproducibility and availability.

## 7 Experiments

For future reference we provide a benchmark of baselines for each task, representation, and data split. We observe that: i) the optimal protein representation differs depending on the task, ii) structure-based splits are harder to generalize to, and iii) pre-training with AlphaFoldDB enhances performance for most models and tasks.

### 7.1 Experimental setup

We consider the following representative deep neural networks for each protein representation: GINs [56] for graphs, PointNet++ [43] for point clouds, and a 3D CNN for voxels. These models serve as our base models for obtaining residue embeddings, which are then either used directly, pooled, or combined with other protein/molecule embeddings to perform predictions at residue level, protein level, or protein pair level, respectively. Note that the voxel model cannot easily be applied in residue-level tasks, as each voxel may contain several residues. Our models are implemented in PyTorch [42] and PyG [16]. For each model, we use minimal hyper-parameter tuning to show proof-of-concept results. More advanced structure-aware machine learning models [19, 9, 28] could potentially boost performances. We cordially invite researchers to contribute their own models to our leaderboard. The data parameters are left to the default settings of `ProteinShake`. Graphs are constructed using an $\epsilon$-neighborhood with $\epsilon = 8.0\,\text{Å}$, the voxel grids are constructed using a voxel size of $2.0\,\text{Å}$ and a $35 \times 35 \times 35$ voxel grid size. We report model size, runtime and memory usage in Appendix Table 6. We refer the interested reader to our code base for the details of model architectures and hyper-parameter settings.

Additionally to supervised training on the respective task training data, we pre-train models on structures from AlphaFoldDB which are provided by `ProteinShake`. For simplicity, we only consider a masking residue strategy, adapted from techniques used in natural language modeling [31]. The masking strategy consists of randomly sampling a set of residues to mask, by replacing the true residue type with a special mask label. Finally, we independently maximize the log likelihood of the

---

[2]With permanent DOI object identifiers issued by Zenodo. See our documentation for a list of releases.

true residue type given the structure and the remaining unmasked residues as context. We add a linear layer on top of the pre-trained base model and finetune the entire model using task-specific losses.[3]

## 7.2 The optimal protein representation differs depending on the task

Table 2 shows a comparison of model performances on different tasks and representations (based on the random split, results for the other splits can be found in the Appendix). While the graph model generally performed the best with large margin, some tasks are better modeled with point clouds or voxel grids.

The tasks describe different physical processes and are heterogeneous in their prediction targets. The data representations on the other hand capture different aspects of protein structure and also the corresponding model architectures process information in very different ways. We hence hypothesize that the different models/representations capture different aspects of proteins which are more or less important for solving a given task.

The tasks that are better solved with point clouds or voxel grids are *Ligand Affinity*, *Protein Protein Interface* and *Structure Similarity*, which heavily rely on spatial, geometric information, which is poorly modeled by a graph. We conclude that researchers should consider a priori which representation best fits their task at hand or include the representation as part of a hyperparameter search.

Table 2: Comparison of models trained with different representations of protein structure across various tasks, on a **random data split**. The optimal choice of representation depends on the task. Shown are mean and standard deviation across four runs with different seeds. The `Voxel` model is not applicable to residue-level tasks.

| Representation
Task | Graph | Point | Voxel |
|---|---|---|---|
| Binding Site | **0.721 ± 0.010** | 0.609 ± 0.006 | - |
| Enzyme Class | **0.790 ± 0.007** | 0.712 ± 0.016 | 0.643 ± 0.026 |
| Gene Ontology | **0.704 ± 0.001** | 0.580 ± 0.002 | 0.602 ± 0.018 |
| Ligand Affinity | 0.670 ± 0.019 | 0.683 ± 0.003 | **0.689 ± 0.013** |
| Protein Family | **0.728 ± 0.004** | 0.609 ± 0.004 | 0.668 ± 0.005 |
| Protein-Protein Interface | 0.883 ± 0.050 | **0.974 ± 0.003** | - |
| Structural Class | **0.495 ± 0.012** | 0.293 ± 0.013 | 0.337 ± 0.011 |
| Structural Similarity | 0.598 ± 0.018 | 0.627 ± 0.006 | **0.645 ± 0.020** |

## 7.3 Generalization is harder in structure-based splits

Figure 3 shows a comparison of data splits, across tasks and representations. Generally, models decrease in performance going from a random split, to a sequence split, to a structure split. This indicates that models struggle to generalize to splits based on sequence and structure similarity. Vice versa, this implies that sequence and structure similarity provide information that models can utilize to improve their predictions.

In a scenario where a model is applied to out-of-distribution (OOD) data it is advisable to construct appropriate train/test splits to correctly assess generalization capability. Most real world applications require a model to work on OOD data. Hence, whenever a model is used to predict properties of proteins that are either dissimilar in sequence or in structure to the train set, we advise to use the corresponding split of `ProteinShake` for evaluation.

The evaluations presented here were split on a moderate 70% similarity threshold, where the effect is already drastic, across tasks and models. We emphasize that real world applications regularly present such distribution shifts. The data split type should be carefully considered, and in case of doubt the most stringent structure split should be used.

---

[3]All code for training and pre-training is available at https://github.com/BorgwardtLab/proteinshake_models.

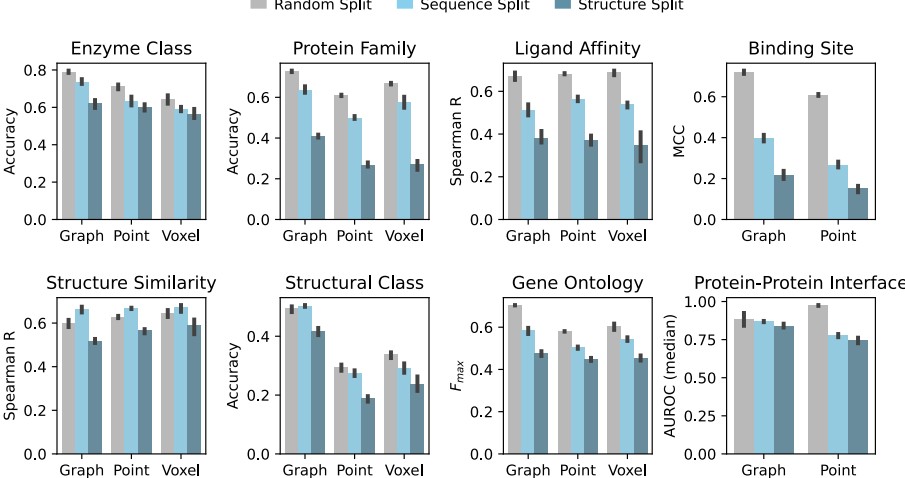

Figure 3: Comparison of random, sequence, and structure splits across tasks and representations. Models generalize less well to sequence and structure splits, respectively. The split should be chosen according to the prediction problem.

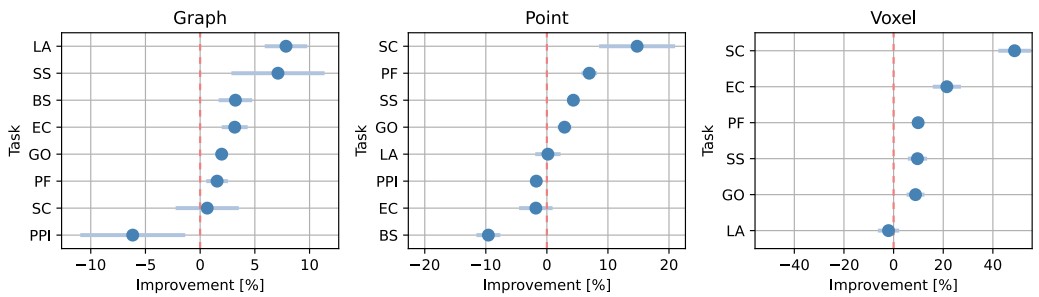

Figure 4: Relative improvement due to pre-training across tasks and representations. In most settings, performance is substantially improved by pre-training with AlphaFoldDB. Tasks are abbreviated with their initials. Values are relative to the metric values obtained from the supervised model without pre-training.

## 7.4 Pre-training with AlphaFoldDB enhances performance across models and tasks

Unsupervised pre-training with unlabeled data has become a standard strategy to improve model performance on supervised tasks [15, 59, 35]. We can also see this effect for our prediction tasks on *experimentally* determined protein structures, even though we used *predicted* structures from AlphaFoldDB for pre-training (Figure 4). Pre-training substantially improves model performance in the majority of test cases, however there are some model-task combinations where initialization with pre-trained weights is even detrimental. This means that, in some cases, the parameters learned under the pre-training objective (here residue masking) are less informative for solving the task than random initializations. The mechanisms of pre-training are still discussed [39, 2, 23], other pre-training objectives might be more appropriate for certain tasks and models. `ProteinShake` aims to simplify further research on topics such as this one by making tasks and model architectures directly comparable.

The availability of high quality structure predictions presents many possibilities for improving performance on supervised tasks. We strongly advise researchers to make use of this data. `ProteinShake` provides the SwissProt database from AlphaFoldDB readily accessible as a pre-processed dataset.

# 8 Conclusion

We presented `ProteinShake`, an open-source Python package which unifies data preparation for machine learning on protein 3D structures, eliminating one of the most tedious and error-prone stages of the workflow for practitioners. The tool is highly flexible while providing rich functionality in a simple, accessible and extensible way. Our experiments demonstrate this flexibility and provided some imperatives for model development. We also showcased the capabilities of the library as a framework for building datasets and benchmarks by implementing a diverse set of tasks, covering many biological research questions and machine learning problems. A limitation of these benchmarks is that they are simplified approximations to complex biological problems, made with inter-task comparability in mind (e.g. a uniform splitting strategy across tasks). Future releases of our benchmarks will include further task-specific considerations and will be complemented with re-implementations of existing benchmarks.

We hope to have laid a foundation for researchers to now integrate both established and novel datasets and benchmarks. Importantly, `ProteinShake` was built explicitly to facilitate community collaboration, we therefore invite researchers to contribute custom datasets, tasks, and leaderboard submissions to the library. We believe `ProteinShake` will improve reproducibility and greatly facilitate machine learning model development in structural protein biology.

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
