# Appendix

`ProteinShake`: Building datasets and benchmarks for deep learning
on protein structures

**Homepage**

proteinshake.ai

**Source code**

github.com/BorgwardtLab/proteinshake

**Documentation**

proteinshake.readthedocs.io/en/latest

**Models**

github.com/BorgwardtLab/proteinshake_models

## Benchmark results on random, sequence and structure split

Table 3: Comparison of models trained with different representations of protein structure across various tasks, on a **random data split**. The optimal choice of representation depends on the task. Shown are mean and standard deviation across four runs with different seeds. The Voxel model is not applicable to residue-level tasks.

| Representation
Task | Graph | Point | Voxel |
|---|---|---|---|
| Binding Site | **0.721 ± 0.010** | 0.609 ± 0.006 | - |
| Enzyme Class | **0.790 ± 0.007** | 0.712 ± 0.016 | 0.643 ± 0.026 |
| Gene Ontology | **0.704 ± 0.001** | 0.580 ± 0.002 | 0.602 ± 0.018 |
| Ligand Affinity | 0.670 ± 0.019 | 0.683 ± 0.003 | **0.689 ± 0.013** |
| Protein Family | **0.728 ± 0.004** | 0.609 ± 0.004 | 0.668 ± 0.005 |
| Protein-Protein Interface | 0.883 ± 0.050 | **0.974 ± 0.003** | - |
| Structural Class | **0.495 ± 0.012** | 0.293 ± 0.013 | 0.337 ± 0.011 |
| Structure Similarity | 0.598 ± 0.018 | 0.627 ± 0.006 | **0.645 ± 0.020** |

Table 4: Comparison of models trained with different representations of protein structure across various tasks, on a **sequence data split**. Shown are mean and standard deviation across four runs with different seeds.

| Representation
Task | Graph | Point | Voxel |
|---|---|---|---|
| Binding Site | **0.399 ± 0.020** | 0.268 ± 0.018 | - |
| Enzyme Class | **0.737 ± 0.016** | 0.635 ± 0.029 | 0.591 ± 0.014 |
| Gene Ontology | **0.582 ± 0.018** | 0.503 ± 0.006 | 0.543 ± 0.010 |
| Ligand Affinity | 0.513 ± 0.031 | **0.565 ± 0.011** | 0.538 ± 0.013 |
| Protein Family | **0.635 ± 0.020** | 0.500 ± 0.008 | 0.577 ± 0.028 |
| Protein-Protein Interface | **0.869 ± 0.005** | 0.776 ± 0.012 | - |
| Structural Class | **0.503 ± 0.004** | 0.275 ± 0.012 | 0.292 ± 0.019 |
| Structure Similarity | 0.663 ± 0.016 | 0.667 ± 0.004 | **0.671 ± 0.020** |

Table 5: Comparison of models trained with different representations of protein structure across various tasks, on a **structure data split**. Shown are mean and standard deviation across four runs with different seeds.

| Representation
Task | Graph | Point | Voxel |
|---|---|---|---|
| Binding Site | **0.219 ± 0.022** | 0.152 ± 0.019 | - |
| Enzyme Class | **0.621 ± 0.026** | 0.600 ± 0.021 | 0.567 ± 0.031 |
| Gene Ontology | **0.474 ± 0.014** | 0.448 ± 0.008 | 0.454 ± 0.014 |
| Ligand Affinity | **0.383 ± 0.034** | 0.374 ± 0.024 | 0.350 ± 0.084 |
| Protein Family | **0.411 ± 0.009** | 0.269 ± 0.012 | 0.270 ± 0.027 |
| Protein-Protein Interface | **0.840 ± 0.016** | 0.747 ± 0.023 | - |
| Structural Class | **0.415 ± 0.015** | 0.188 ± 0.011 | 0.236 ± 0.027 |
| Structure Similarity | 0.518 ± 0.010 | 0.564 ± 0.011 | **0.587 ± 0.039** |

## Details of the models used in the experiments

Table 6: Number of parameters, runtime in terms of proteins processed per second (Proteins/s), and memory consumption for each model and task pair. The throughput, adapted from the computer vision field [48], is measured as the number of proteins that we can process per second on one 10GB H100 GPU MIG. For each model we take the largest possible batch size (bs) and calculate the average time over 30 runs to process that batch. The memory consumption is calculated with the Pytorch.profiler library.

| Task | Graph (GIN) | Point (PointNet++) | Voxel (Conv3d) |
|---|---|---|---|
| Binding Site | 1.3M | 0.6M | - |
| Enzyme Class | 1.3M | 0.6M | 7.2M |
| Gene Ontology | 2.6M | 1.3M | 8.5M |
| Ligand Affinity | 1.8M | 0.9M | 7.7M |
| Protein Family | 2.7M | 1.3M | 8.5M |
| Protein-Protein Interface | 1.5M | 0.7M | - |
| Structure Class | 2.1M | 1.1M | 8.0M |
| Structure Similarity | 1.5M | 0.7M | 7.4M |
| Proteins/s GPU (H100 10G,fp32) | 1.4K | 1.3K | 83 |
| Proteins/s CPU (bs=1) | 75 | 42 | 2 |
| Memory GPU (bs=32,fp32) | 1.71GB | 0.8GB | 14.4GB |