# OpenReview forum: "ProteinShake: Building datasets and benchmarks for deep learning on protein structures"
_NeurIPS.cc/2023/Track/Datasets_and_Benchmarks — NeurIPS 2023 Datasets and Benchmarks Poster_

### Official Review · Reviewer_XEHS · 2023-06-30

**Rating:** 8
**Confidence:** 4

**Strengths:**

## Significance and relevance to the broader research community
Machine Learning for biology is a subfield of increasing importance. One of the limiting factors for ML researchers to enter the field is  that considerable domain-expertise is necessary to construct meaningful datasets and tasks. Providing benchmarks like the one proposed here removes some of these barriers and is thus potentially very valuable to the field.

## quality of the research
The approach taken by the authors is systematic and seems well thought trough. The authors set up a series of criteria for benchmarks/datasets to be useful to the community: amenability, availability, transparency, extendibility and compatibility, and address these systematically in the design of their software library.  Generally, the tasks seem meaningful, and care has been taken to use relevant splits for training/test.

## clarity of paper
The paper is very clearly written and supported by high quality figures.

## Accessibility, Accountability, and transparency
My impression from the paper and website is that care has been taken to ensure accessibility and transparency. The authors claim that the database will be "regularly updated", and provides code for releasing new versions of the datasets, suggesting that this is more than an empty promise.

**Additional Feedback:**

## Very minor details:

Line 18, extra space after "algorithms"

Line 162 *"See Figure 3"*
This reference is on page 5, while the figure is on page 9. Consider moving it a bit closer.

**Clarity:**

The paper is very well written. Please see comments about "clarity" under "Strengths" above.

**Correctness:**

The experiments described in the paper are sound, and the conclusions drawn appear well supported by experiments.

**Documentation:**

Although the benchmark is generally well motivated and well designed, there were several instances where technical details were lacking, which I've pointed out in the comments above. The paper would benefit from a more elaborate supporting information document containing such details (or with a more elaborate website - which would make it possible to update the information if other tasks/experiments were added in the future).

**Ethics:**

I have no ethical concerns about the contents of this paper.

**Limitations:**

The authors have not explicitly addressed limitations of their proposed benchmark set.

**Opportunities For Improvement:**

I appreciate the idea of writing up a list of desiderata prior to the discussion of the design of the dataset itself. One potential suggestion to the authors is to consider whether "Reproducibility" should be added as an additional criterion. Clearly, for a benchmark to be useful it is critical that experiments can be run against exactly the same time-stamped version of the datasets (to avoid the need of having to rerun already results published in the literature). The authors seem to have considered this, and towards the end of the paper, they mention "fully automated, versioned releases" (line 216), but it might be fruitful to make this part of the guiding principles in the beginning of the paper.

Line 104. It was not clear to me exactly which annotations were provided. Are the amino acid identities included? Are the b-factor/pLDDT included? It would make sense if this was spelled out either in an appendix (or on the website), so that users could quickly determine if the data contained the information they need.

Line 112. *"There are two large-scale (unannotated) datasets..."*
Related to my previous question, does this mean that amino acid identities and b-factor/pLDDT are not provided for these large datasets? Both would be very useful: amino acid identities for inverse-folding type analysis, pLDDT for determining the trustworthiness of input structures.

Line 135. It was not clear to me how structure information was encoded into voxels. What are the channels? And is some smoothing applied (to obtained superresolution). Would be nice if such details were described in the Supporting Material (or on the website).

Line 150. *"clustered by sequence similarity"*
At which level of homology reduction? (it says 70% somewhere later in the paper, which seems high). Since the community does not always agree about the proper level of homology reduction, how easy is it as a user to change this value? I assume that this decision is baked into the "versioned datasets", so would it make sense to include a few predefined levels of homology reduction in each version?

Line 158. *"For any protein in the test set, all similar proteins above this threshold are moved to the test set, ensuring an upper bound on the similarity between train and test proteins."*
Is this done recursively (do you check again for the new proteins added)? - otherwise I don't see that you are ensured an upper bound.

Line 166. Section on Structure-function relationships.
I was surprised that you included no tasks on sequence-structure relationships. I understand why you might want to avoid including a 3D structure prediction task (since we already have CASP), but in recent years we have seen quite some work on predicting amino acid identities from structural contexts (sometimes referred to as inverse folding). Since this information is in your dataset, it would be great to see this task included (especially because there has been some debates about which representations are most useful for these tasks).

Line 202. ProteinProteinInterfaceTask. Unclear exactly what the input information is. Are the two protein structures provided in unbound form?

Line 204. BindingSiteDetectionTask. Similar question as above. Exactly which data is provided? A structure and an index into the structure?

Line 206. VirtualScreenTask. Are the small molecules provided as SMILES?

Line 229. *"Note that the voxel model cannot easily be applied in residue-level tasks, as each voxel may contain several residues."*
This is not a fundamental limitation - since voxels sizes can be chosen to precluded occupancy by several residues.

Line 236. *"size of 10.0 Å and a 10 × 10 × 10"*
This is a very coarse grid, which I assume will put the CNN at quite a disadvantage. For a fair comparison to the graph representation, it might have made sense to use a higher resolution grid for the experiments in the paper.

Line 276 When using pre-trained models, it is not quite clear what the architecture is of the supervised layer on top. Please clarify.

Figure 4 caption. *"Performance is substantially improved by pre-training with AlphaFoldDB."*
From the figure, it was not obvious that this holds generally. For several cases it seems neutral and in some cases even negative. Am I misreading the plots?

**Relation To Prior Work:**

To my knowledge, the manuscript provides a fair representation of existing work in the field.

**Summary And Contributions:**

The manuscript introduces a software package to facilitate the training of machine learning models on protein structure. It provides both datasets, tasks and an API that allows for easy experimentation with different protein structure representations. The focus is on ease of use, while simultaneously allowing users to amend the library with additional data and tasks.

---

> ### Author Response · Authors · 2023-08-22
> **Part I**
>
> > I appreciate the idea of writing up a list of desiderata prior to the discussion of the design of the dataset itself. One potential suggestion to the authors is to consider whether "Reproducibility" should be added as an additional criterion. Clearly, for a benchmark to be useful it is critical that experiments can be run against exactly the same time-stamped version of the datasets (to avoid the need of having to rerun already results published in the literature). The authors seem to have considered this, and towards the end of the paper, they mention "fully automated, versioned releases" (line 216), but it might be fruitful to make this part of the guiding principles in the beginning of the paper.
>
> Thank you for this suggestion. As we use the word "reproducible" in another context before, we have added a new desideratum "recency" to explicitly mention the need for versioning and keeping datasets up to date (Section 2).
>
> > It was not clear to me exactly which annotations were provided. Are the amino acid identities included? Are the b-factor/pLDDT included? It would make sense if this was spelled out either in an appendix (or on the website), so that users could quickly determine if the data contained the information they need.
> > "There are two large-scale (unannotated) datasets..." Related to my previous question, does this mean that amino acid identities and b-factor/pLDDT are not provided for these large datasets? Both would be very useful: amino acid identities for inverse-folding type analysis, pLDDT for determining the trustworthiness of input structures.
>
> We see that we have passed over this detail and have added it to the manuscript (Section 4), thank you for the note. Amino acid identities are included, as well as pLDDT values. Other features may easily be added as well.
> The [documentation for the Dataset class](https://proteinshake.readthedocs.io/en/latest/modules/datasets.html) also contains a table with all the default attributes.
>
> > It was not clear to me how structure information was encoded into voxels. What are the channels? And is some smoothing applied (to obtained superresolution). Would be nice if such details were described in the Supporting Material (or on the website).
>
> We have added these details to the [documentation](https://proteinshake.readthedocs.io/en/latest/modules/representations.html#proteinshake.representations.VoxelDataset). Briefly, the channels are one-hot encoded amino acid identities aggregated per voxel via a user-defined aggregation function (default: mean). Superresolution is not yet implemented, but would be an interesting addition to the library.
>
> > "clustered by sequence similarity" At which level of homology reduction? (it says 70% somewhere later in the paper, which seems high). Since the community does not always agree about the proper level of homology reduction, how easy is it as a user to change this value? I assume that this decision is baked into the "versioned datasets", so would it make sense to include a few predefined levels of homology reduction in each version?
>
> We have added some more information to the manuscript (Section 5.1). We pre-compute splits for every dataset at several homology levels ranging from 50% to 90%, which can be easily accessed by changing a [function argument](https://proteinshake.readthedocs.io/en/latest/modules/tasks.html#:~:text=sequence%E2%80%99%2C%20or%20%E2%80%98structure%E2%80%99.-,split_similarity_threshold,-(float)%20%E2%80%93%20Maximum) of the task. Full control is given when pre-processing the datasets locally.
>
> > "For any protein in the test set, all similar proteins above this threshold are moved to the test set, ensuring an upper bound on the similarity between train and test proteins." Is this done recursively (do you check again for the new proteins added)? - otherwise I don't see that you are ensured an upper bound.
>
> This was poor phrasing and we have removed the statement, thank you for spotting it. Indeed one cannot guarantee an upper bound without removing proteins at the "decision boundary". The similarity thresholds are merely defined on cluster representatives.

---

> > ### Author Response · Authors · 2023-08-22
> > **Part II**
> >
> > > Section on Structure-function relationships. I was surprised that you included no tasks on sequence-structure relationships. I understand why you might want to avoid including a 3D structure prediction task (since we already have CASP), but in recent years we have seen quite some work on predicting amino acid identities from structural contexts (sometimes referred to as inverse folding). Since this information is in your dataset, it would be great to see this task included (especially because there has been some debates about which representations are most useful for these tasks).
> >
> > This is a very timely and interesting task, in fact this task is currently under development for upcoming releases of ProteinShake. For this submission however we started with a small set of representative datasets to showcase some capabilities and design choices of the library. In a way our pre-training objective does something similar to inverse folding, as we predict the identity of a masked amino acid.
> >
> > > ProteinProteinInterfaceTask. Unclear exactly what the input information is. Are the two protein structures provided in unbound form?
> > > BindingSiteDetectionTask. Similar question as above. Exactly which data is provided? A structure and an index into the structure?
> > > VirtualScreenTask. Are the small molecules provided as SMILES?
> >
> > Thank you for pointing this out, we have added more information in the manuscript (Section 5.2.3) and [documentation](https://proteinshake.readthedocs.io/en/latest/modules/tasks.html).
> > Briefly:
> >
> > * ProteinProteinInterface task gives the model two chains that come from a bound complex. We mimick the unbound state by centering and randomly rotating the two chains before prediction. The task is then to predict all chain-chain contacts.
> > * BindingSiteDetectionTask we label the residues and atoms of each protein by noting which of them are found within a small molecule binding pocket (as defined by PDBBind). Only the protein is given to the model which classifies the residues as belonging to a binding pocket or not. This is analogous to the problem of [pocket detection](https://doi.org/10.1186/1471-2105-10-168) in drug discovery.
> > * VirtualScreenTask: Small molecules are indeed provided as SMILES strings, we have amended this in the manuscript.
> >
> >
> > > "Note that the voxel model cannot easily be applied in residue-level tasks, as each voxel may contain several residues." This is not a fundamental limitation - since voxels sizes can be chosen to precluded occupancy by several residues.
> >
> > That is true, however the voxel size is a choice of the user, who might choose to use a larger voxel size. Hence we did not want to suggest evaluation and comparison on this subtask.
> >
> > > "size of 10.0 Å and a 10 × 10 × 10" This is a very coarse grid, which I assume will put the CNN at quite a disadvantage. For a fair comparison to the graph representation, it might have made sense to use a higher resolution grid for the experiments in the paper.
> >
> > We chose this parameter to keep model sizes comparable between representations. As it was also raised by another reviewer, we have decided to re-run the voxel model with a higher resolution (voxel size=2Å). We ask for understanding that we can only provide updated values for Table 2 at this time, and will deliver the remaining results in the camera-ready version when model (pre-)training has finished.
> >
> > > When using pre-trained models, it is not quite clear what the architecture is of the supervised layer on top. Please clarify.
> >
> > Thank you for raising this issue, we have included these details in Section 7.1. Specifically, we used a [TaskHead](https://github.com/BorgwardtLab/proteinshake_models/blob/220aa24c76dab88b1e4d46f812acf6a2ec34fb79/proteinshake_eval/models/protein_model.py#L44) module which provides a task-specific aggregator and an output layer. For the output layer, we used a linear layer. We train both the pretrained backbone model and the TaskHead for fine-tuning. All our code for training and pre-training is available [here](https://github.com/BorgwardtLab/proteinshake_models).

---

> > > ### Author Response · Authors · 2023-08-22
> > > **Part III**
> > >
> > > > Figure 4 caption. "Performance is substantially improved by pre-training with AlphaFoldDB." From the figure, it was not obvious that this holds generally. For several cases it seems neutral and in some cases even negative. Am I misreading the plots?
> > >
> > > No on the contrary, we agree that our discussion of this plot was a bit shallow. We have rephrased this paragraph (Section 7.4) and note that pre-training only has a positive effect in some cases, in others it is even detrimental. The effect of pre-training is an under-explored topic in the field, and we cannot provide further insight on why this happens. However, we hope that ProteinShake provides the means to further study different pre-training strategies and their effects.
> > >
> > > > The authors have not explicitly addressed limitations of their proposed benchmark set.
> > >
> > > We have added a brief discussion of our limitations in Section 8. Namely, that unifying long-standing and disparate biological tasks under one umbrella requires many technical choices which we are still in the process of improving for future data releases.
> > >
> > > > Although the benchmark is generally well motivated and well designed, there were several instances where technical details were lacking, which I've pointed out in the comments above. The paper would benefit from a more elaborate supporting information document containing such details (or with a more elaborate website - which would make it possible to update the information if other tasks/experiments were added in the future).
> > >
> > > Thank you for raising this concern, we have added some more information in the manuscript (more details on all tasks, more discussion on data splitting in Section 5.1 and on data attributes in Section 4). Our documentation provides a more detailed and up-to-date description with references to related work and thorough explanation of the API which we could not fit into the manuscript.

---

> > > > ### Comment · Reviewer_XEHS · 2023-08-23
> > > > **Response to rebuttal**
> > > >
> > > > Thanks to the authors for a detailed response to my comments and questions. The authors have addressed the minor concerns I had, and I will update my score to an 8.
> > > >
> > > > My only additional suggestion is that the authors perhaps consider defining homology reduced splits at even lower levels than 50%. A common rule of thumb is that protein sequences are considered homologues if they have more than 30% sequence identity.

---

> > > > > ### Author Response · Authors · 2023-08-25
> > > > >
> > > > > Thank you, much appreciated!
> > > > >
> > > > > PS: We have added the 30% split and will update the manuscript with the next revision.

---

### Official Review · Reviewer_8GvG · 2023-07-03
**Building datasets and benchmarks for deep learning on protein structures**

**Rating:** 6
**Confidence:** 4

**Strengths:**

+ Allow users to use three different representations of protein structures and provide some guidance for users to choose different representations for different tasks
+ The datasets are in the format ready for being used by some standard deep learning packages.
+ Three ways of splitting datasets are useful.
+ Clustering proteins to create splits is useful.
+ Including the model evaluation into the pipeline is valuable.
+ Pretraining models on AlphaFold predicted structures is very useful.

**Additional Feedback:**

N/A

**Clarity:**

The manuscript is well written and easy to understand. However, some details are missing (see the comments for opportunities to improve for details).

**Correctness:**

The results and description in this work are generally correct. However, some choice may be questionable. For instance, the size of voxel is set to 10 Angstrom, which seems too coarse. Why not use a smaller voxel size such as 1, 2, or 3 Angstrom to get better resolution?

**Documentation:**

The web site does not provide secured access. Therefore, the documentation is not evaluated.

**Ethics:**

There is no ethics concern.

**Limitations:**

-	Users cannot add standard test datasets in the field into the system to control the generation of the training datasets such that the similarity between the training datasets and the standard test datasets is below a threshold.
-	Different prediction tasks may need different evaluation metrics. However, there is little discussion about the evaluation metrics. Some evaluation metrics such as AUROC for protein interface prediction task may not be appropriate because of the highly imbalance distribution of the positive and negative classes. The protein interface prediction task describe in this work is a very challenging task. But the  AUROC-mean score (0.974) reported in this work is very high, probably because it does not consider how well the method can predict the positive cases.
-	Some prediction tasks may be ill defined. For instance, predicting if a pair of residues from two proteins are in their interaction interface is very similar to the inter-protein contact prediction widely used in the field. However, the relationship between the two is not discussed. The latter appears to be better defined and easier to understand. The practical value of another task (StructureSearchTask (retrieval): retrieve a set of proteins structurally similar to a query) is not articulated provided that FoldSeek can quickly retrieve a set of proteins structurally similar to a query.
-	This website https://borgwardtlab.github.io/proteinshake/#leaderboard does not provide secured access.
-	There is lack of rationale why freesasa was chosen over other tools such as DSSP to calculate solvent accessibility scores.

**Opportunities For Improvement:**

•	Provide some detailed description about how the datasets have been created, how their quality and completeness are ensured, and what their advantages are in comparison with the standard datasets for the prediction tasks.
•	Provide some detailed description about how users may create their own datasets using the software package.
•	Provide more description about how users can use the model evaluation modules to evaluate their own results using the model evaluation metrics. More information about the evaluation metrics for the prediction tasks and their justification are needed.
•	Provide function to allow users to integrate the pretrained model with their models.
•	Allow users to add the standard test datasets into the data split process.
•	Provide function for users to remove / control redundancy (similar proteins) within test or training datasets.

**Relation To Prior Work:**

The relation with the prior work about the software packages/frameworks for creating protein datasets is well described. However, the relation to the standard datasets for different prediction tasks is not well discussed.

**Summary And Contributions:**

This manuscript reports a software package ProteinShake for users to create protein structure datasets to train deep learning models for various prediction tasks and evaluate the performance of the models. 10 datasets for different tasks created by the software package are provided. The test results of the in-house methods on the datasets are presented. The three methods of splitting the data (random, sequence similarity-based, and structure similarity-based) are useful. The function of generating three representations (graph, voxel, and point cloud) from protein structures for deep learning is useful. Preparing the data in the format suitable for standard deep learning packages such as pyTorch and TensorFlow makes using them much easy. Applying in-house methods on the test datasets show they can be valuable for some protein structure prediction tasks. The work also shows that pretraining models on AlphaFold predicted structures is useful for improving the downstream prediction tasks and provides the pretraining capability for users. Overall, the idea and direction of standardizing protein data creation and model evaluation are very good, and if successfully implemented, they can significantly advance the application of deep learning in this field.

However, it is not clear how the datasets are curated to ensure their completeness and good quality. There is little description about how users may use the software package to create their own datasets for the tasks described in this manuscript or new protein prediction tasks. There some widely used datasets (e.g., CAFA test datasets for protein function prediction) for many tasks described in this work, but there is no comparison between the datasets curated in this work and the existing ones to lay out their advantages or complementarity. Some prediction tasks (i.e., predict if a pair of residues are in interaction interface of two proteins) are not well defined and are similar to the existing and even better-defined tasks (e.g., inter-protein residue-residue contact prediction). And there is no option for users to include standard tests into the framework to control the data split. Even though the useful function of clustering proteins is provided, there is no option for users to remove / control redundancy (highly similar proteins) in either test or training datasets. It is not clear how users can use the model evaluation module to evaluate the results of their methods. And there is lack of description about how users can integrate the model pretrained on AlphaFold predicted structures with their models.

---

> ### Author Response · Authors · 2023-08-22
> **Part I**
>
> We thank the reviewer for this highly constructive feedback. First we would like to apologize for the inconvenience of our documentation security settings, we had some temporary configuration issues with routing on the webserver. The issue has been resolved and the documentation is accessible now through HTTPS. If there are any further problems with access please let us know and we will promtly address it. We kindly ask to briefly re-evaluate the documentation at this time, as it provides answers and clarifications to several of the raised items.
>
> We grouped some of the comments to be able to address them together. As such, the order of comments has changed.
>
> > However, it is not clear how the datasets are curated to ensure their completeness and good quality.
>
> > Provide some detailed description about how the datasets have been created, how their quality and completeness are ensured, and what their advantages are in comparison with the standard datasets for the prediction tasks.
>
> Where possible, we link the respective paper introducing the data, and, if applicable, a representative method paper that uses the data in a prediction problem. Briefly, datasets are created mostly from the RCSB PDB database and annotations stored there, or from databases dedicated to a certain prediction problem, such as PDBbind for protein binding and interaction. Quality control is at this time only minimal and we rely mostly on the curation efforts of the primary databases, but ProteinShake provides the functionality to filter proteins on all annotations and a thorough quality control can easily be extended in the future. For comparison to existing datasets, please see a comment further below.
>
> > There is little description about how users may use the software package to create their own datasets for the tasks described in this manuscript or new protein prediction tasks.
>
> > Provide some detailed description about how users may create their own datasets using the software package.
>
> Our documentation provides a [tutorial](https://proteinshake.readthedocs.io/en/latest/notes/custom.html) on how to create custom datasets and tasks.

---

> > ### Author Response · Authors · 2023-08-22
> > **Part II**
> >
> > > There some widely used datasets (e.g., CAFA test datasets for protein function prediction) for many tasks described in this work, but there is no comparison between the datasets curated in this work and the existing ones to lay out their advantages or complementarity.
> >
> > > Some prediction tasks (i.e., predict if a pair of residues are in interaction interface of two proteins) are not well defined and are similar to the existing and even better-defined tasks (e.g., inter-protein residue-residue contact prediction).
> >
> > > And there is no option for users to include standard tests into the framework to control the data split.
> >
> > > Allow users to add the standard test datasets into the data split process.
> >
> > > Users cannot add standard test datasets in the field into the system to control the generation of the training datasets such that the similarity between the training datasets and the standard test datasets is below a threshold.
> >
> > > Some prediction tasks may be ill defined. For instance, predicting if a pair of residues from two proteins are in their interaction interface is very similar to the inter-protein contact prediction widely used in the field. However, the relationship between the two is not discussed. The latter appears to be better defined and easier to understand.
> >
> > > The relation with the prior work about the software packages/frameworks for creating protein datasets is well described. However, the relation to the standard datasets for different prediction tasks is not well discussed.
> >
> > We implemented a handful of datasets and tasks to cover a broad range of biological applications and well as machine learning problems. The aim was to showcase that the library can accomodate different types of tasks. As such, these datasets and tasks may be simplified or phrased differently than established datasets. We have added more information and references for each of the datasets/tasks in the manuscript (Section 5.2) and [documentation](https://proteinshake.readthedocs.io/en/latest/modules/tasks.html) to make this clearer. Among others, we have added a clarification to the `ProteinProteinInterfaceTask` to better position it with the literature. Our intention was indeed to frame it as inter-chain contact prediction, and some text was added to reflect this (Section 5.2.3).
> >
> > ProteinShake is however easily extended to include standard benchmarks. To demonstrate, we have added a [tutorial](https://proteinshake.readthedocs.io/en/latest/notes/custom.html#custom-splits) on how to add the CAFA benchmark in our documentation. This tutorial also details how custom test splits can be used and integrated from existing work.
> >
> > We want to note that some of these established benchmarks, specifically CAFA, were designed with a different goal in mind. We expressly focus on structure as the primary datatype. In CAFA, the existence of a structure is not a requirement and many test proteins can therefore not be used in a structure model.
> >
> > > Even though the useful function of clustering proteins is provided, there is no option for users to remove/control redundancy (highly similar proteins) in either test or training datasets.
> >
> > > Provide function for users to remove / control redundancy (similar proteins) within test or training datasets.
> >
> > We apologize, this feature was a bit hidden in our documentation. We have added it explicitly to the manuscript (Section 5.1). We pre-compute all splits with similarity thresholds ranging from 50% to 90%, which can be selected by simply changing a [function argument](https://proteinshake.readthedocs.io/en/latest/modules/tasks.html#:~:text=sequence%E2%80%99%2C%20or%20%E2%80%98structure%E2%80%99.-,split_similarity_threshold,-(float)%20%E2%80%93%20Maximum).
> >
> > > It is not clear how users can use the model evaluation module to evaluate the results of their methods.
> >
> > > Provide more description about how users can use the model evaluation modules to evaluate their own results using the model evaluation metrics.
> >
> > The documentation provides [examples](https://proteinshake.readthedocs.io/en/latest/notes/quickstart.html) for all representations and frameworks.

---

> > > ### Author Response · Authors · 2023-08-22
> > > **Part III**
> > >
> > > > More information about the evaluation metrics for the prediction tasks and their justification are needed.
> > >
> > > > Different prediction tasks may need different evaluation metrics. However, there is little discussion about the evaluation metrics. Some evaluation metrics such as AUROC for protein interface prediction task may not be appropriate because of the highly imbalance distribution of the positive and negative classes. The protein interface prediction task describe in this work is a very challenging task. But the AUROC-mean score (0.974) reported in this work is very high, probably because it does not consider how well the method can predict the positive cases.
> > >
> > > We have added more information and references about the metrics in the manuscript (Section 5.2), and also link to the respective papers in the [documentation](https://proteinshake.readthedocs.io/en/latest/modules/tasks.html). We should note that we implement several metrics for each task, and only report a default metric in the manuscript. This default metric is taken from the literature where possible.
> > > In the particular case you mentioned we agree that the imbalanced problem calls for an approprate metric, and that AUROC most certainly overestimates the performance. We hence provide imbalanced metrics in the code, but report our results as others have done (Townshend et. al 2021, Neurips Datasets and Benchmarks).
> > >
> > > > And there is lack of description about how users can integrate the model pretrained on AlphaFold predicted structures with their models.
> > >
> > > > Provide function to allow users to integrate the pretrained model with their models.
> > >
> > > The code for our baselines and instructions on how to use them are located in a [separate repository](https://github.com/BorgwardtLab/proteinshake_models). We have added there a [code example](https://github.com/BorgwardtLab/proteinshake_models#using-pretrained-models-with-custom-prediction-heads) on how to use the pre-trained models with custom prediction heads.
> > >
> > > > The practical value of another task (StructureSearchTask (retrieval): retrieve a set of proteins structurally similar to a query) is not articulated provided that FoldSeek can quickly retrieve a set of proteins structurally similar to a query.
> > >
> > > FoldSeek is indeed a very strong contestant for this task, but others may still wish to approach it, possibly comparing to FoldSeek.
> > > In general, the problem of protein structure similarity computation is known to have NP-Hard instantiations (Goldman et al. 1999, IEEE, "Algorithmic aspects of protein structure similarity."), and current methods often operate on various heuristics. We believe it is likely that future works in this direction will benefit from a common benchmark.
> > >
> > > > There is lack of rationale why freesasa was chosen over other tools such as DSSP to calculate solvent accessibility scores.
> > >
> > > This was merely a convenience choice, but other attributes and features may be easily integrated. We are already considering DSSP for secondary structure features and might use it for SASA as well in the future.
> > >
> > > > The results and description in this work are generally correct. However, some choice may be questionable. For instance, the size of voxel is set to 10 Angstrom, which seems too coarse. Why not use a smaller voxel size such as 1, 2, or 3 Angstrom to get better resolution?
> > >
> > > We are currently re-running the voxel models with a smaller voxel size. We provide the results of Table 2 with updated values. As model training (especially pre-training) takes some time, we ask for understanding that we will provide the remaining results in the camera-ready version.

---

> > > > ### Comment · Reviewer_8GvG · 2023-08-29
> > > >
> > > > The authors addressed most of my comments. I increased my score to 6.

---

### Official Review · Reviewer_j1GE · 2023-07-22
**Easy to use library making protein data available to the ML community**

**Rating:** 7
**Confidence:** 3
**Correctness:** The dataset/task design seems correct…

**Strengths:**

- The paper proposes tasks for models with different objectives, modelling structure-function relationships, geometric protein structure and physical interactions. Similarly, the paper offers different splits to reflect real-world settings and to adjust the difficulty of the task in terms of generalization.
- The pipeline seems easy to work with and is accessible with many different frameworks (e.g., PyTorch/TensorFlow, PyG/DGL, etc.).


**Additional Feedback:**

.

**Clarity:**

- The paper is well written and the contributions are clearly stated.
- Different effects of pretraining in Figure 4 could be discussed in more detail as there are tasks with strong positive and negative impact.


**Documentation:**

- Source code is fully available and datasets are hosted including a leaderboard.
- It would be interesting to add more statistics about the datasets, e.g., protein size, etc.

**Limitations:**

This work is a great step towards enabling a larger community to work with protein data. Naturally, also due to the relatively short conference format, it cannot replace a rigorous study of the subject, which might be in many cases necessary to make substantial progress.

**Opportunities For Improvement:**

- The benchmarking aspect could benefit from adding further baseline models (at least 2 per modality), to show if the proposed tasks and datasets are suitable to differentiate different models’ performances.
- The benchmark study solely focusses on the model’s performance. The model’s time/space requirement could be a good addition, e.g., reporting seconds per protein.
- The empirical results could be accompanied with some more analysis. High-level explanations of the biochemical context of different tasks is given, but it is hard to assess the actual relevance and especially the difficulty of a task. In that context, it would be interesting to add further information about the datasets, e.g., avg. protein size, avg. degree of graphs, etc., and to analyze how this affects performance.


**Relation To Prior Work:**

Related work is sufficiently discussed.

**Summary And Contributions:**

- The paper proposes an end-to-end pipeline that facilitates applying machine learning models to protein structures.
- The paper provides 8 annotated and 2 unannotated datasets.
- The proposed tasks are grouped in 3 groups, guiding model training towards specific domain-specific objectives.
- Proteins can be represented in 3 ways: point cloud, graph or voxel grid, which can be advantageous depending on the modality.
- The paper shows pretraining often improves performance.

---

> ### Author Response · Authors · 2023-08-22
>
> > The benchmarking aspect could benefit from adding further baseline models (at least 2 per modality), to show if the proposed tasks and datasets are suitable to differentiate different models’ performances.
>
> In this submission we wanted to demonstrate the capabilities of the library by providing some baselines. Since we evaluate them in many settings, this is already a considerable effort. We hope that models contributed from the community will assess the benchmarks more thoroughly in the future. We will monitor the submissions and adjust the tasks if necessary.
>
> > The benchmark study solely focusses on the model’s performance. The model’s time/space requirement could be a good addition, e.g., reporting seconds per protein.
>
> Thank you for this suggestion, we have added these metrics to our models (Appendix Table 6) and will also include them in the leaderboard for future submissions.
>
> > The empirical results could be accompanied with some more analysis. High-level explanations of the biochemical context of different tasks is given, but it is hard to assess the actual relevance and especially the difficulty of a task. In that context, it would be interesting to add further information about the datasets, e.g., avg. protein size, avg. degree of graphs, etc., and to analyze how this affects performance.
>
> Thank you for this suggestion. Some general statistics about each task can be found in the [documentation](https://proteinshake.readthedocs.io/en/latest/notes/overview.html), which we have extended according to your suggestions.
>
> > This work is a great step towards enabling a larger community to work with protein data. Naturally, also due to the relatively short conference format, it cannot replace a rigorous study of the subject, which might be in many cases necessary to make substantial progress.
>
> Thank you, we do hope that ProteinShake will enable a larger community to compare the same problems across different models and modalities. While we cannot provide a rigorous study on all of these subjects, we hope to provide the foundation such that others can collectively study these problems in depth.
>
> > Different effects of pretraining in Figure 4 could be discussed in more detail as there are tasks with strong positive and negative impact.
>
> We have rephrased this paragraph (Section 7.4), it is however notoriously difficult to investigate the inner workings of pre-training. The field is still discussing why and how pre-training works. It is interesting to note though that one can see these differences in models and tasks with ProteinShake, which is what we aim to provide. We hope that this way others can study the effect of pre-training in more detail.
>
> > It would be interesting to add more statistics about the datasets, e.g., protein size, etc.
>
> We have [added a few more statistics](https://proteinshake.readthedocs.io/en/latest/notes/overview.html#statistics), if you have more suggestions, please let us know!

---

### Official Review · Reviewer_Da4b · 2023-07-26
**comprehensive and timely addition to help further deep learning on protein tasks**

**Rating:** 9
**Confidence:** 3
**Correctness:** Yes
**Clarity:** Yes

**Strengths:**

Major strengths of this work are: 1) the ability to split data using either sequence or structure-based similarity; the latter is particularly important for structure-based tasks and is often not used by those in the field. 2) The generation and benchmarking of three different ways of representing protein structures (graphs, point clouds, or voxels). 3) Well-described benchmarks on popular tasks.

**Additional Feedback:**

NA

**Documentation:**

yes

**Ethics:**

no issues

**Limitations:**

Yes

**Opportunities For Improvement:**

Did the authors consider including/pre-training on the ESM Metagenomic Atlas? It's my understanding it's much larger than the AlphafoldDB.

**Relation To Prior Work:**

yes

**Summary And Contributions:**

This paper describes a software package that is designed to simplify and standardize dataset loading processing and curation specifically for deep learning workflows. The major contributions here are 1) the ability to load many different existing datasets already preprocessed in different representations tailored for structure-based deep learning architectures; and 2) benchmarks on how these different representations perform in various tasks. This package looks very well-reasoned and executed. The paper is written clearly for experts and those newish to either DL or protein structure area.

---

> ### Author Response · Authors · 2023-08-22
>
> Thank you for your positive feedback. Using ESM structure data is indeed a good suggestion. We did consider this possibility, but working with that much data poses some challenges, both to the performance of the library as well as to the computing infrastructure. We are actively working on solving these, and hope to provide also this dataset in the coming weeks.

---

> > ### Comment · Reviewer_Da4b · 2023-08-29
> > **Response to rebuttal**
> >
> > I look forward to the addition of the ESM dataset and others as well.

---

### Author Response · Authors · 2023-08-22
**Thank you to reviewers and global response**

We begin with a sincere thank you to all reviewers for their thoughtful reviews and overall positive comments.

It was encouraging to see that the reviewers valued the key features of our library, such as the different representations and splits, or ease of use and a diverse set of tasks.

The common concerns across reviewers were clarifications regarding the definition of the tasks and technical choices during dataset construction. We address these points individually in direct replies. Generally, we have made the following adjustments in the current revision:

* The construction of tasks and datasets with respect to existing baselines in the literature was clarified and supported with further references (Section 5.2)
* More code examples and tutorials have been added to the documentation, in particular a [tutorial](https://proteinshake.readthedocs.io/en/latest/notes/custom.html#custom-splits) for implementing established benchmarks such as CAFA.
* Each task has a [Task Summary](https://proteinshake.readthedocs.io/en/latest/modules/tasks.html) which details in plain language the input/output and evaluation metrics for each task.
* The documentation includes more [dataset statistics](https://proteinshake.readthedocs.io/en/latest/notes/overview.html).
* We are currently conducting experiments with smaller voxel sizes. New results with voxel size 2 Angstroms were added to Table 2 (see table below for direct comparison to previous setting). Runtime and memory usage for the 3 model classes are reported in Appendix Table 6.


Comparison of Voxel size

| Task | Voxel (2A)   | Voxel (10A)  |
|:---- | ------------ | ------------ |
| BS   | -            | -            |
| EC   | 0.643+-0.026 | 0.656+-0.012 |
| GO   | 0.602+-0.018 | 0.609+-0.004 |
| LA   | 0.689+-0.013 | 0.690+-0.015 |
| PF   | 0.668+-0.005 | 0.543+-0.007 |
| PP   | -            | -            |
| SC   | 0.337+-0.011 | 0.221+-0.014 |
| SS   | 0.645+-0.020 | 0.620+-0.010 |




Changes to the text are highlighted in red in the updated manuscript. Responses to all other points are placed in the individual answers to each reviewer below.

Once again, we thank the reviewers for their high quality feedback which we feel has substantially improved the submission.

We are happy to address further questions and comments throughout the discussion period.

---

### Decision · Program_Chairs · 2023-09-22

**Decision:**

Accept (Poster)

**Comment:**

The reviewers are generally enthusiastic about this work and consider this dataset and benchmark make it easy for the community to develop machine learning methods to study proteins. The concerns raised by the reviewers were well addressed by the authors during the rebuttal period.